# TOKENIZE IMAGE AS A SET

## ABSTRACT

This paper proposes a new paradigm for image generation through set-based to-kenization and modeling. Unlike conventional methods that serialize images into fixed-position latent codes with a uniform compression ratio, we introduce an un-ordered token set representation to dynamically allocate coding capacity based on regional semantic complexity. This TokenSet enhances global context aggregation and improves robustness against local perturbations. To address the critical chal-lenge of modeling discrete sets, we devise a dual transformation mechanism that bijectively converts sets into fixed-length integer sequences while preserving sum-mation constraints. Further, we propose Fixed-Sum Discrete Diffusion—the first framework to simultaneously handle discrete values, fixed sequence length, and summation invariance—enabling effective set distribution modeling. Experiments demonstrate our method's superiority in semantic-aware representation and gener-ation quality. Our innovations, spanning novel representation and modeling strate-gies, advance visual generation beyond traditional sequential token paradigms.

## 1 INTRODUCTION

Contemporary visual generation frameworks (Esser et al., 2021; Sun et al., 2024; Li et al., 2024a; Yu et al., 2024) predominantly adopt a two-stage paradigm: first compressing visual signals into latent representations, then modeling the low-dimensional distributions. Conventional tokenization methods (Kingma et al., 2013; van den Oord et al., 2017; Esser et al., 2021; Yu et al., 2022) typi-cally employ uniform spatial compression ratios, generating serialized codes with fixed positional correspondence. Consider a beach photo where the upper half is a sky region that contains min-imal detail; the lower half contains a semantically dense foreground—current approaches allocate the same number of codes to both regions. This raises a fundamental question: Should visually simplistic regions receive the same representational capacity as semantically rich areas?

This paper introduces a novel way of visual com-pression and distribution modeling, *TokenSet* (Fig-ure 1). During the compression stage, we pro-pose to tokenize images into unordered sets rather than position-dependent sequences. Unlike serial-ized tokens that maintain fixed spatial correspon-dence, our token set enables dynamic attention allo-cation based on regional semantic complexity. This approach enhances global information aggregation, facilitates semantic-aware representation, and ex-hibits superior robustness to local perturbations.

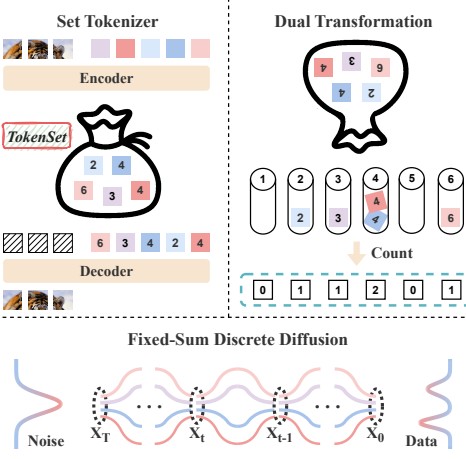

Figure 1: Pipeline of our method.

Nevertheless, modeling set-structured data presents significant challenges compared to sequential coun-terparts. Existing set modeling approaches fall into two categories: The first class (Zaheer et al., 2017; Lee et al., 2019; Edwards & Storkey, 2016) adopts pooling-based operations (e.g., mean/sum/max/similar operations) to extract low-dimensional features, it lacks direct supervi-sion on each element in the set, often yielding suboptimal results; Other correspondence-based meth-ods (e.g., DETR's Hungarian matching (Carion et al., 2020)) seek to construct element-wise supervi-

sion through bipartite matching. However, the inherent instability of dynamic matching mechanisms causes supervisory signals to vary across training iterations, leading to suboptimal convergence.

To address it, we devise a *dual transformation* mechanism that converts set modeling into a sequence modeling problem. Specifically, we count the occurrences of each unique token index in the set, transforming unordered data into a structured sequence where: (i) the sequence length equals the codebook size, (ii) each element represents non-negative integer counts, (iii) the summation of all elements equals the number of elements in the set.

While existing discrete modeling approaches handle fixed-length integer sequences but ignore summation constraints, and continuous diffusion models can preserve element sums while struggling with discrete value representations, no current approach simultaneously satisfies all three constraints. We therefore propose Fixed-sum Discrete Diffusion. By introducing a constant-sum prior, we simultaneously satisfy all three critical properties and achieve effective modeling of this structured data.

Our contributions can thus be summarized as:

1. We propose a novel set-based image tokenization method that departs from serialized representations by exhibiting global contextual awareness, enabling dynamic token allocation based on semantic complexity while maintaining robustness against local perturbations.

2. We propose an effective solution for modeling discrete set data through dual transformation, establishing a bijection between unordered sets and serialized data.

3. Fixed-sum Discrete Diffusion, a dedicated generative framework that explicitly enforces summation constraints in discrete data modeling, achieves superior modeling of set distribution.

## 2 RELATED WORK

### 2.1 IMAGE TOKENIZATION

Image tokenization compresses images from high-dimensional pixel space into a compact representation, facilitating subsequent understanding and generation tasks. Early approaches like Variational Autoencoders (VAEs) (Kingma et al., 2013) map input images into low-dimensional continuous latent distributions. Building on this, VQVAE (van den Oord et al., 2017) and VQGAN (Esser et al., 2021) project images into discrete token sequences, associating each image patch with an explicit discrete token. Subsequent works VQVAE-2 (Razavi et al., 2019), RQVAE (Lee et al., 2022), and MoVQ (Zheng et al., 2022) leverage residual quantization strategies to encode images into hierarchical latent representations. Meanwhile, FSQ (Mentzer et al., 2023), SimVQ (Zhu et al., 2024b), and VQGAN-LC (Zhu et al., 2024a) address the representation collapse problem when scaling up codebook sizes. Other innovations include a dynamic quantization in DQVAE (Huang et al., 2023), integration of semantic information in ImageFolder (Li et al., 2024b), and architectural refinements (Yu et al., 2021; Cao et al., 2023). Recently, TiTok (Yu et al., 2024) explores 1D latent sequences for image representation, achieving good reconstruction at an impressive compression ratio.

Despite these advances, previous approaches predominantly encode images into token sequences, where each element corresponds to fixed image positions. This paper proposes representing images as unordered token sets, thereby eliminating positional bias while capturing global visual semantics.

### 2.2 SET MODELING

Early set-based representations include Bag-of-Words (BoW) (Salton et al., 1975; Joachims, 1998; Pang et al., 2002) and its visual counterparts (Sivic & Zisserman, 2003; Csurka et al., 2004; Lazebnik et al., 2006). More recently, CoC (Ma et al., 2023) proposes to treat an image as a set of points via clustering. However, these set-based representations lose some information from the original data. Conversely, certain data modalities—such as point clouds and bounding boxes—inherently align with set representations. This has motivated substantial research efforts to model permutation-invariant data, yet it presents three fundamental challenges.

Firstly, prevailing generative paradigms such as auto-regressive (AR) models (Sun et al., 2024; Tian et al., 2025; Esser et al., 2021) and diffusion models (Ho et al., 2020; Dhariwal & Nichol, 2021;

Gu et al., 2022; Gat et al., 2025; Peebles & Xie, 2023) are designed for sequential data modeling, making them incompatible with unordered set-structured data.

Secondly, processing permutation-invariant data necessitates strictly symmetric operations (e.g., sum, max, or similar) to avoid positional dependencies (Zaheer et al., 2017; Lee et al., 2019; Edwards & Storkey, 2016). However, this constraint prevents the use of powerful tools such as convolution and attention, thereby bottlenecking the model's capacity.

Thirdly, effective modeling of complex data distributions typically requires per-element supervision signals, yet unordered sets inherently lack such mechanisms. Existing approaches like DSPN (Zhang et al., 2019) employ Chamfer loss for supervision, while TSPN (Kosiorek et al., 2020) and DETR (Carion et al., 2020) utilize Hungarian matching. However, these matching processes are inherently unstable and often lead to inconsistent training signals. Alternative approaches like PointCloudGAN (Li et al., 2018) attempt to directly model the global distribution, which compromises training effectiveness and overall performance.

This paper bypasses these limitations through a dual transformation operation, effectively transforming sets into sequences. This transformation enables us to leverage various sequence-based modeling methods to tackle the challenging task of set modeling.

## 3 METHOD

### 3.1 IMAGE SET TOKENIZER

The key to tokenizing an image into a set is to eliminate position dependencies between visual tokens and the fixed position of the image. While prior work TiTok (Yu et al., 2024) converts images into 1D token sequences by removing 2D positional relationships, it preserves fixed 1D positional correspondence. We start from this approach and develop a completely position-agnostic tokenization framework. Our architecture employs Vision Transformers (ViT) (Dosovitskiy et al., 2021) for both encoder and decoder. The encoder processes image patches

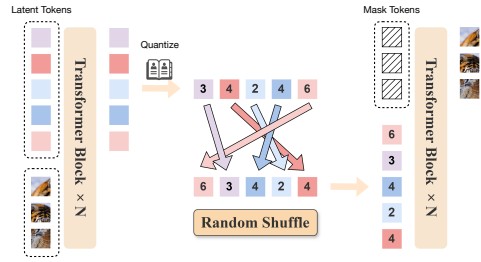

Figure 2: Pipeline of our set tokenizer.

alongside learnable latent tokens, producing latent representations that are discretized through a VQVAE (van den Oord et al., 2017) codebook. This process generates a 1D token sequence $\mathbf{T} = [t_1, t_2, \ldots, t_M]$, where $t_i \in \{0, 1, \ldots, C-1\}$, with $C$ denoting the codebook size and $M$ representing the token count. To eliminate the 1D position bias, we introduce permutation invariance during decoding. Specifically, we define $\mathcal{T}$ as the set representation of $\mathbf{T}$ [1]:

$$\mathcal{T} = \{t_1, t_2, \ldots, t_M\}, \tag{1}$$

where all permutations of $\mathbf{T}$ are considered equivalent. During training, we randomly permute tokens before decoder input while maintaining reconstruction targets, as illustrated in Figure 2. Although the permutation space grows factorially with $M$, empirical results demonstrate effective learning of permutation invariance through partial permutation sampling.

This set-based tokenization demonstrates three principal advantages over serialized tokenization: First, by decoupling tokens from fixed spatial positions, the model learns to dynamically allocate tokens based on global image content rather than local patch statistics. Second, the global receptive field significantly improves noise robustness by preventing over-reliance on local features. Third, through training, tokens spontaneously develop specialized attention patterns focusing on semantically distinct regions. We empirically verify these characteristics in Section 4.2.

### 3.2 DUAL TRANSFORMATION

After tokenizing, an image is represented as an unordered token set $\mathcal{T} = \{t_1, t_2, \ldots, t_M\}$. Modeling such complex sets using neural networks presents significant challenges, primarily due to the inherent unordered nature of sets and the lack of effective supervision of individual elements.

---

[1]Strictly speaking, the structure should be referred to multiset due to the inclusion of duplicate elements.

Existing sequential modeling approaches, particularly autoregressive (Esser et al., 2021) and diffusion models (Dhariwal & Nichol, 2021; Gu et al., 2022), face inherent limitations when processing set-structured data. These methods fundamentally rely on the positional ordering of elements, making them not suitable for permutation-invariant sets where both element order ambiguity and exponential permutation possibilities exist. Alternative approaches like PointGAN (Li et al., 2018) suffer from training instability and a lack of efficient representation for permutation invariant data. Other methods like DETR (Carion et al., 2020) leverage Hungarian matching to achieve the set correspondence. However, it suffers from matching instability, hindering robust modeling.

To address these challenges, we propose a dual transformation mechanism (Figure 1) that bidirectionally converts between unordered sets and structured sequences. Given a token set $\mathcal{T} = \{t_1, t_2, \ldots, t_M\}$, we construct a count vector $\mathbf{X} = (x^0, x^1, \ldots x^{C-1}) \in \mathbb{N}^C$ through:

$$x^j = \sum_{i=1}^{M} \delta(t_i, j), \quad \text{for } j = 0, 1, \ldots, C-1, \tag{2}$$

where $\delta(t_i, j)$ denotes the Kronecker delta function. By doing this, we effectively convert the unordered token set into serialized data without loss of any information. Furthermore, the converted sequence data $\mathbf{X}$ has three critical structural priors:

- **Fixed-length sequence**: The count vector $\mathbf{X}$ contains $C$ elements, corresponding to the size of the codebook, ensuring a fixed length sequence.
- **Discrete count values**: Each element $x_j$, which records codebook item frequencies, is an integer in $[0, M]$, where $M$ is the number of tokens extracted from the encoder.
- **Fixed-sum constraint**: The summation of all values equals the number of encoded tokens.

In summary, the dual transformation establishes a bidirectional mapping between set and sequence representations, offering two fundamental advantages: (1) It reduces the challenging problem of modeling permutation-invariant sets to the well-studied domain of sequence modeling, crucially enabling auto-regressive and diffusion frameworks for modeling. (2) The identified structural priors—fixed sequence length, discrete value constraints, and summation conservation—provide mathematically grounded regularization that guides effective model learning.

### 3.3 FIXED-SUM DISCRETE DIFFUSION

Given that our dual-transformed sequential data exhibits three prior properties, we investigated several modeling approaches. While both auto-regressive models (Sun et al., 2024) and standard discrete diffusion methods (Gu et al., 2022) are effective for discrete-valued data, with the latter being particularly suitable for fixed-length sequences, they do not inherently guarantee the fixed-summation property. Conversely, continuous diffusion models (Peebles & Xie, 2023) can naturally preserve both fixed-length and fixed-summation constraints through their mean-preserving MSE loss (Lin et al., 2024), but they struggle with discrete distribution modeling (Chen et al., 2022).

To synergistically combine the strengths of these approaches while satisfying all three priors, we propose a novel modeling approach called Fixed-Sum Discrete Diffusion (FSDD), illustrated in Figure 3. This method integrates a constrained diffusion path within a discrete flow matching architecture, inspired by continuous diffusion methods that enforce summation constraints during intermediate denoising steps. The key innovation lies in ensuring that samples at every intermediate step strictly adhere to the fixed-sum constraint.

#### 3.3.1 TRAINING PIPELINE

**Diffusion Process**: The initial noise sample $\mathbf{X}_1$ is drawn from a multinomial distribution over integer vectors of length $C$ with a fixed summation $M$:

$$\mathbf{X}_1 = (x_1^0, x_1^1, \ldots, x_1^{C-1}), \text{where} \quad 0 \leq x_1^j \leq M, \quad \sum_{j=0}^{C-1} x_1^j = M. \tag{3}$$

Given $\mathbf{X}_1$ sampled from the noise distribution and $\mathbf{X}_0$ from the data distribution, both satisfying $\sum \mathbf{X} = M$, we define the constrained diffusion process as:

$$q(\tilde{\mathbf{X}}_t | \mathbf{X}_1, \mathbf{X}_0) = \mathcal{N}(\boldsymbol{\mu}_t, \boldsymbol{\sigma}_t^2). \tag{4}$$

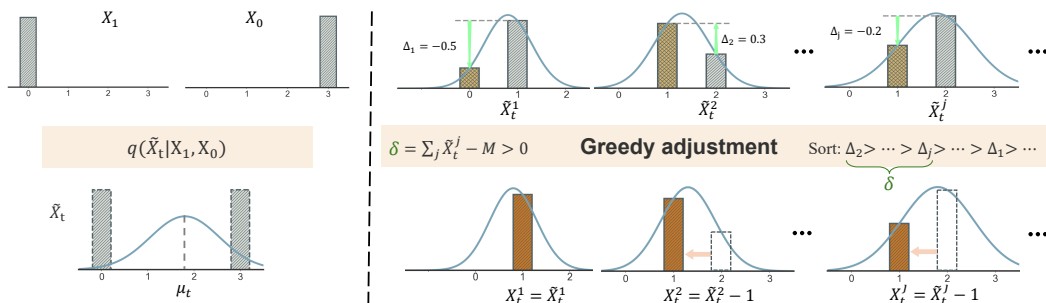

Figure 3: Fixed-Sum Diffusion Process. Sample $X_t$ at noise level $t = 0.6$ is first sampled from the mixed gaussian distribution of $X_0$ and $X_1$, and then adjusted through greedy adjustment. Samples dropped during greedy adjustment are marked with dashed line.

The parameters $\boldsymbol{\mu}_t$ and $\boldsymbol{\sigma}_t$ satisfy:

$$\boldsymbol{\mu}_t = t\,\mathbf{X}_1 + (1-t)\,\mathbf{X}_0, \quad \boldsymbol{\sigma}_t = \frac{|\mathbf{X}_1 - \mathbf{X}_0|}{4}. \tag{5}$$

This design guarantees the constraint on the summation expectation:

$$\mathbb{E}_{\tilde{\mathbf{X}}_t}\left[\sum \tilde{\mathbf{X}}_t\right] = M, \quad \forall\, t \in [0,1]. \tag{6}$$

However, we cannot guarantee that each individual sample $\tilde{\mathbf{X}}_t$ satisfies this constraint. Therefore, we perform dynamic adjustments to ensure it.

**Greedy Adjusting**: Our adjustment protocol operates under the core objective of preserving the likelihood of $\tilde{\mathbf{X}}_t$. Specifically, if the summation of $\tilde{\mathbf{X}}_t$ exceeds $M$, we reduce certain elements of $\tilde{\mathbf{X}}_t$ through a greedy selection criterion: For each element, we quantify the likelihood degradation caused by each adjustment and prioritize adjusting those that can increase the likelihood or minimize its reduction. Thus, the adjusted sample $\mathbf{X}_t$ adheres to both the fixed-sum constraints and the probability distribution in Equation (4). We provide an illustration in

---

**Algorithm 1** Greedy Adjusting Sampling

---

**Require:** Target sum $M$, sample distribution $q$
**Ensure:** Sample $\mathbf{X}_t$ satisfying $\sum \mathbf{X}_t = M$
1: Sample $\tilde{\mathbf{X}}_t \sim q(\cdot|\mathbf{X}_1, \mathbf{X}_0)$
2: $\delta \leftarrow \sum \tilde{\mathbf{X}}_t - M$
3: $\mathbf{d} \leftarrow q(\tilde{\mathbf{X}}_t - \mathrm{sgn}(\delta)\mathbf{1}|\mathbf{X}_1, \mathbf{X}_0) - q(\tilde{\mathbf{X}}_t|\mathbf{X}_1, \mathbf{X}_0)$

4: $\mathbf{j}^* \leftarrow \arg\mathrm{Topk}(\mathbf{d}, |\delta|)$
5: Initialize $\mathbf{X}_t \leftarrow \tilde{\mathbf{X}}_t$
6: Adjust $\mathbf{X}_t[\mathbf{j}^*] \leftarrow \mathbf{X}_t[\mathbf{j}^*] - \mathrm{sgn}(\delta)\mathbf{1}[\mathbf{j}^*]$
7: **return** $\mathbf{X}_t$

---

Figure 3 and pseudo code in Algorithm 1. During training, we implement this greedy adjustment strategy at every diffusion step to integrate the fixed-sum constraint.

Moreover, the fixed-sum discrete diffusion employs the standard discrete diffusion loss, where the denoising network $\theta$ is trained via cross-entropy loss to predict $\mathbf{X}_0$ from noisy input $\mathbf{X}_t$, maintaining discrete state transitions while preserving the summation invariant.

### 3.3.2 INFERENCE STRATEGY

The inference process of Fixed-Sum Diffusion follows an iterative denoising scheme with enforced summation constraints. Starting from a noise sample $\mathbf{X}_1$ that satisfies $\sum \mathbf{X}_1 = M$, we progressively refine the sample through discrete transitions:

$$p_\theta(\mathbf{X}_{t-\Delta t}|\mathbf{X}_t) = \sum_{\mathbf{x}_0} q(\mathbf{X}_{t-\Delta t}|\mathbf{X}_t, \mathbf{X}_0)\, p_\theta(\mathbf{X}_0|\mathbf{X}_t), \tag{7}$$

Here, $p_\theta(\mathbf{X}_0|\mathbf{X}_t)$ represents the discrete data distribution predicted from the noisy data. We employ the top-$p$ sampling strategy to generate $\mathbf{X}_0$ candidates, which are then processed through the posterior term $q(\tilde{\mathbf{X}}_{t-\Delta t}|\mathbf{X}_t, \mathbf{X}_0)$. This term implements a truncated Gaussian discretization:

$$q(\tilde{\mathbf{X}}_{t-\Delta t}|\mathbf{X}_t, \mathbf{X}_0) = \mathcal{N}(\boldsymbol{\mu}_{t-\Delta t}, \boldsymbol{\sigma}^2_{t-\Delta t}), \tag{8}$$

with parameters defined as:

$$\boldsymbol{\mu}_{t-\Delta t} = \left(1 - \frac{\Delta t}{t}\right)\mathbf{X}_t + \frac{\Delta t}{t}\mathbf{X}_0, \quad \boldsymbol{\sigma}_{t-\Delta t} = \frac{|\mathbf{X}_1 - \mathbf{X}_0|}{4} \cdot f(t - \Delta t), \tag{9}$$

where $f(\cdot)$ controls the truncation ratio during sampling. To ensure strict adherence to the summation constraint, we apply the greedy adjustment on $\tilde{\mathbf{X}}_{t-\Delta t}$ to ensure $\sum \mathbf{X}_{t-\Delta t} = M$, effectively bridging the potential gap between the training and inference phases.

## 4 EXPERIMENTS

### 4.1 SETTING

We conducted our experiments on the ImageNet dataset (Deng et al., 2009), with images at a resolution of $256 \times 256$. We report our results on the 50,000-images ImageNet validation set, utilizing the Fréchet Inception Distance (FID) metric (Heusel et al., 2017). Our evaluation protocol is provided by (Dhariwal & Nichol, 2021).

**Implementation Details.** For tokenizer training, we followed the strategy in TiTok (Yu et al., 2024) and applied data augmentations, including random cropping and horizontal flipping. We used the AdamW optimizer (Loshchilov & Hutter, 2019) with a base learning rate of $1e\text{-}4$ and a weight decay of $1e\text{-}4$. The model is trained on ImageNet for $1000k$ steps, with a batch size of 256, equivalent to 200 epochs. We implemented a learning rate warm-up phase followed by a cosine decay schedule, with gradient clipping at a threshold of $1.0$. An Exponential Moving Average (EMA) with a 0.999 decay rate was adopted, and we report results from the EMA models. To enhance quality and stabilize training, we incorporated a discriminator loss (Esser et al., 2021) and trained only the decoder during the final $500k$ steps. Additionally, we utilized MaskGIT's proxy code (Chang et al., 2022) following Yu et al. (2024) to facilitate training.

The generator configuration aligned with DiT (Peebles & Xie, 2023). We used random horizontal flipping as data augmentation. All models were optimized with AdamW (Loshchilov & Hutter, 2019) using a constant learning rate of 1e-4 and a batch size of 256, and trained for 200 epochs. We implemented EMA with a decay rate of 0.9999 throughout the training. For inference, we utilized 25 sampling steps combined with classifier-free guidance to further enhance the image quality.

### 4.2 SET TOKENIZER

In contrast to sequential image tokenization approaches, representing images as token sets introduces distinct properties, including permutation invariance, global context awareness, and enhanced robustness against local perturbations. Furthermore, we demonstrate that set-based tokenization can simultaneously achieve precise reconstruction while inherently organizing tokens into semantically coherent clusters.

#### 4.2.1 PERMUTATION-INVARIANCE

We test the permutation-invariance of our tokenizer by reconstructing images from encoded tokens arranged in different orders. Specifically, we decode the tokens in five different sequence orders: (1) the original order, (2) the reversed order, (3) a randomly shuffled order, (4) tokens sorted in ascending order, and (5) tokens sorted in descending order. As shown in Figure 4, all reconstructed images are visually identical, indicating the permutation-invariance of our tokenizer. This invariance is quantitatively corroborated by a stable rFID of 3.62 across all tested orders. These findings demonstrate that the network can successfully learn permutation invariance by training on only a subset of permutations.

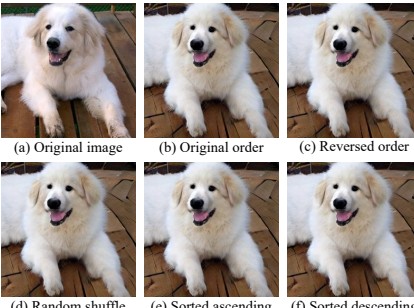

(a) Original image    (b) Original order    (c) Reversed order

(d) Random shuffle    (e) Sorted ascending    (f) Sorted descending

Figure 4: Visual comparison of the reconstructed images from various order permutations of the encoded tokens. All reconstructed images are identical.

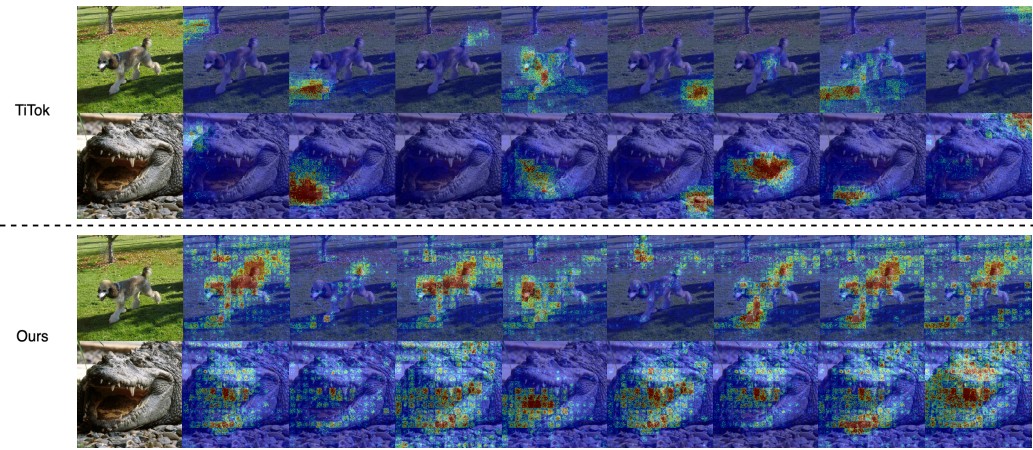

Figure 5: Comparison of token receptive fields between our method and TiTok (Yu et al., 2024).

Table 1: Robustness analysis of different tokenizers. We report the percentage (%) of overlapping tokens between those generated from the original and the noise-altered images. Although our method is set-based, we provide results obtained by treating token sets as sequences (marked by †).

| Method | #Tokens | Signal-to-Noise Ratio (dB) | | | | |
|---|---|---|---|---|---|---|
| | | 40 | 30 | 20 | 10 | 1 |
| VQGAN | 256 | 69.2 | 36.6 | 10.9 | 1.6 | 0.6 |
| TiTok | 128 | 83.4 | 55.4 | 21.4 | 3.8 | 1.4 |
| TiTok | 256 | 77.2 | 44.5 | 13.2 | 1.8 | 0.7 |
| Ours † | 128 | 89.4 | 68.1 | 38.1 | 12.7 | 6.8 |
| Ours † | 256 | 87.0 | 62.4 | 31.1 | 9.0 | 4.3 |
| Ours | 128 | **89.5** | **68.4** | **38.9** | **13.9** | **7.9** |
| Ours | 256 | 87.6 | 64.0 | 33.2 | 10.6 | 5.7 |

Table 2: Linear probing results on ImageNet validation set. We list the reported results of a strong self-supervised method MAE (He et al., 2022) for reference. † denotes MAE trained for the same 200 epochs as our tokenizer.

| Method | Encoder | #Tokens | #Codebook | Acc@1 |
|---|---|---|---|---|
| MAE† | ViT-L | - | - | 64.4 |
| MAE | | | | 75.1 |
| Ours | ViT-B | 128 | 1024 | 44.8 |
| | | 128 | 2048 | 43.1 |
| | | 128 | 4096 | 59.7 |
| | | 128 | 8192 | 61.0 |
| | | 32 | 4096 | **66.2** |
| | | 64 | 4096 | 64.9 |
| | | 128 | 4096 | 59.7 |
| | | 256 | 4096 | 47.2 |

### 4.2.2 GLOBAL CONTEXT AWARENESS

By enforcing permutation invariance, our framework decouples inter-token positional relationships, thereby eliminating sequence-induced spatial biases inherent in conventional image tokenization. This architectural design encourages each token to holistically integrate global contextual information. To empirically validate this phenomenon, we visualize the effective receptive field in Figure 5. Notably, traditional sequence-based tokenizers, such as TiTok, exhibit tight spatial coupling between tokens and fixed local regions. Conversely, our approach fundamentally eliminates positional bias and represents images through the composition of tokens with global receptive fields.

### 4.2.3 ROBUSTNESS

Our set tokens, unbound to specific spatial positions yet capturing global image semantics, demonstrate enhanced robustness to noise. Table 1 compares the robustness of different tokenizers against Gaussian noise injected into input images. Specifically, we added Gaussian noise with varying standard deviations to images and measured the token overlap ratio between the perturbed images and the original ones. The results indicate that our tokenizer consistently achieves higher overlap ratios across all noise levels compared to other tokenizers, such as TiTok and VQGAN. Furthermore, while all methods experience performance degradation with increasing noise intensity, the degradation in our approach occurs at a slower rate, underscoring its superior robustness to noise.

### 4.2.4 SEMANTIC CLUSTERING

Given $M$ tokens drawn from $C$ classes, the representation space of our set-based tokenizer is $\binom{M+C-1}{C-1}$, significantly smaller than that of a sequence-based tokenizer with a size of $C^M$. The

Table 3: Reconstruction performance on the ImageNet benchmark at $256 \times 256$ resolution.

| Method | #Tokens | #Codebook | rFID↓ |
|---|---|---|---|
| Taming-VQGAN (Esser et al., 2021) | 256 | 16384 | 4.98 |
| RQVAE (Lee et al., 2022) | 256 | 16384 | 3.20 |
| MaskGit-VQGAN (Chang et al., 2022) | 256 | 1024 | 2.28 |
| ViT-VQGAN (Yu et al., 2022) | 1028 | 8192 | 1.28 |
| TiTok (Yu et al., 2024) | 64 | 4096 | 1.70 |
| TokenSet | 128 | 2048 | 3.62 |
| TokenSet | 128 | 4096 | 2.74 |

compact representational space produces a more efficient depiction of the image space. Figure 6 illustrates images whose encoded tokens contain certain classes. Intriguingly, we observe that these token distributions inherently exhibit semantically coherent clustering patterns. For example, images containing six tokens belonging to the 65th class consistently depict birds, while twelve 162nd tokens represent dogs. Furthermore, we verify its semantic clustering capacity through linear probing. The results shown in Table 2 indicate that our tokenizer achieves promising performance.

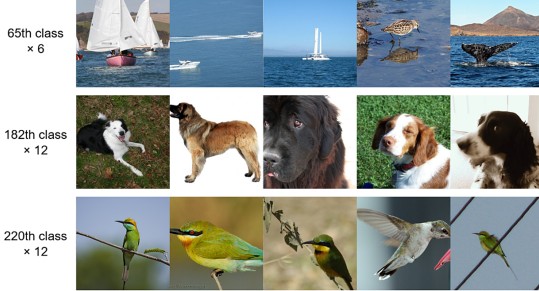

Figure 6: Visualization of images whose encoded tokens share multiple specific classes.

### 4.2.5 RECONSTRUCTION QUALITY

In Table 3, we compare the reconstruction performance of different tokenizers using the ImageNet validation set. Although the random shuffle in our method prevents the network from leveraging the inductive positional bias of images, and the benefits from the representation space of the set are drastically reduced, we find that this seemingly infeasible approach can still achieve good reconstruction performance comparable to previous mainstream methods (Esser et al., 2021; Lee et al., 2022).

### 4.3 FIXED-SUM DISCRETE DIFFUSION

#### 4.3.1 MODELING THROUGH PRIORS

In Table 4a, we compare different modeling methods that leverage different priors. First, as described in Section 4.2.1, any permutation of the image token set can equivalently reconstruct the image. We therefore consider randomly sampling two different permutations for autoregressive modeling, termed AR-order1 and AR-order2. We find that both achieve nearly identical performance. This observation suggests that the permutation-invariant property remains underutilized. To address this, we propose training a autoregressive model to simultaneously learn all possible permutations of the set, termed AR-random. However, this method exhibits poor generation performance due to the difficulty of modeling the large permutation space.

To overcome this limitation, we apply the dual transformation and subsequently model the resulting sequence distribution. We experiment with both autoregressive and discrete diffusion, leveraging set and discrete properties, which perform better than the AR-random. However, they fail to guarantee adherence to our crucial fixed-sum prior, leading to mediocre results. Likewise, modeling discrete distributions by applying continuous diffusion and quantization is also proved ineffective.

In contrast, our Fixed-Sum Discrete Diffusion (FSDD) approach is uniquely capable of simultaneously satisfying all these desired properties, leading to the best performance. This finding demonstrates the importance of incorporating all the requisite priors into the modeling method. It validates our design and highlights the synergy between tokenizer requirements and modeling approach.

#### 4.3.2 ABLATION STUDIES

Prior studies (Esser et al., 2021; Li et al., 2024b) have identified a critical dilemma within the two-stage "compress-then-model" framework for image generation: Increasing the latent space capacity steadily improves reconstruction quality, but generation quality first improves and then declines. In

Table 4: Ablation studies. (a) Ablation on different modeling methods. (b) Impact of token numbers and codebook sizes on reconstruction and generation performance.

(a) Modeling methods.

| Method | Set | Discrete | Fixed-sum | gFID↓ |
|---|---|---|---|---|
| AR-order1 | | ✓ | | 6.55 |
| AR-order2 | | ✓ | | 6.62 |
| AR-random | ✓ | ✓ | | 8.99 |
| SetAR | ✓ | ✓ | | 6.92 |
| Discrete Diffusion | ✓ | ✓ | | 6.23 |
| Continue Diffusion | ✓ | | ✓ | 75.45 |
| FSDD | ✓ | ✓ | ✓ | **5.56** |

(b) Tokens and codebook size.

| #Tokens | #Codebook | rFID↓ | gFID↓ |
|---|---|---|---|
| 128 | 1024 | 6.51 | 9.93 |
| 128 | 2048 | 3.62 | 7.12 |
| 128 | 4096 | 2.74 | **5.56** |
| 128 | 8192 | **2.35** | 8.76 |
| 32 | 4096 | 5.54 | 6.91 |
| 64 | 4096 | 3.54 | 6.03 |
| 128 | 4096 | 2.74 | **5.56** |
| 256 | 4096 | **2.60** | 7.07 |
| 512 | 4096 | 2.88 | 9.37 |

Table 5: Modeling performance comparison on the ImageNet benchmark at $256 \times 256$ resolution.

| Method | #Tokens | #Codebook | #Params | gFID↓ |
|---|---|---|---|---|
| VQGAN (Esser et al., 2021) | 256 | 1024 | 1.4B | 15.78 |
| VQ-Diffusion (Gu et al., 2022) | 1024 | 2886 | 370M | 11.89 |
| MaskGiT (Chang et al., 2022) | 256 | 1024 | 227M | 6.18 |
| LlamaGen (Sun et al., 2024) | 256 | 16384 | 111M | 5.46 |
| LlamaGen (Sun et al., 2024) | 256 | 16384 | 3.1B | 2.18 |
| TiTok (Yu et al., 2024) | 128 | 4096 | 287M | 1.97 |
| TokenSet-S | 128 | 4096 | 36M | 5.56 |
| TokenSet-B | 128 | 4096 | 137M | 5.09 |

this work, the dual transformation converts the compressed distribution into a more tractable form for subsequent modeling, thereby enabling us to rigorously investigate whether this approach can resolve the aforementioned dilemma. We systematically vary the number of tokens and codebook size to study their effects on both reconstruction and generation performance. For generation evaluation, we adopt a small-scale model (36M parameters) to fit the distribution. As illustrated in Table 4b, we find that moderately increasing the latent dimensions enhances both reconstruction and generation quality, but exceeding this range degrades both metrics. This arises from the absence of fixed spatial correspondence between the tokenized latent space and the image grid. Consequently, the decoder cannot establish a direct mapping mechanism to decode latent variables, paralleling the difficulties encountered by diffusion models in modeling excessively intricate latent distributions. Crucially, these observations suggest a potential solution to the reconstruction-generation dilemma: *by removing the decoder's reliance on low-effort shortcut mappings, we can align its behavior more closely with the distribution learning process of the second-stage modeling.*

Our analysis further reveals that scaling the model size yields consistent performance gains in distribution modeling tasks, as empirically demonstrated in Table 5. While these results suggest potential benefits from further model expansion, practical constraints limited our exploration beyond the current experimental scope. We leave this for future work.

## 5 CONCLUSION

This work challenges the conventional paradigm of serialized visual representation by introducing *TokenSet*, a set-based framework that dynamically allocates representational capacity to semantically diverse image regions. Through dual transformation, we establish a bijective mapping between unordered token sets and structured integer sequences, enabling effective modeling of set distributions via our proposed fixed-sum discrete diffusion. Experiments demonstrate that this approach not only achieves dynamic token allocation aligned with regional complexity, but also enhances robustness against local perturbations. By enforcing summation constraints during both training and inference, our framework resolves critical limitations in existing discrete diffusion models while outperforming fixed-length sequence baselines.

Tokenizing image as a set offers distinct advantages over conventional sequential tokenization, introducing novel possibilities for both image representation and generation. This paradigm shift inspires new perspectives on developing next-generation generative models. In future work, we plan to conduct a rigorous analysis to unlock the full potential of this representation and modeling approach.

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
