# OpenReview forum: "Tokenize Image as a Set"
_ICLR.cc/2026/Conference — Submitted to ICLR 2026_

### Official Review · Reviewer_UHKS · 2025-10-22

**Soundness:** 2
**Presentation:** 3
**Contribution:** 3
**Rating:** 4
**Confidence:** 3

**Summary:**

This paper introduces an image tokenizer which is approximately invariant to token order. The authors empirically observe that this leads to tokenized representations that are even more decoupled from spatial/local bias common in other tokenizers (even in TiTok, to some extent, despite its 1D representation).

The authors then introduce “Fixed-Sum Discrete Diffusion” (FSDD), a diffusion-based generative model over multisets/bags of tokens, in order to leverage the tokenizer’s latent space for image generation.

**Strengths:**

While the method used to encourage this approximate permutation invariance is very simple (random token shuffling during training), results demonstrating semantically aligned token receptive fields, high linear probing accuracy, and robustness to noise in the input, are interesting and motivate the importance of such a seemingly minor tweak to TiTok.

The “Fixed-Sum Discrete Diffusion” approach is novel and is demonstrated to outperform certain baselines which do not exploit the permutation invariance.

**Weaknesses:**

Generation performance appears to be compromised compared to TiTok (FID ~5.1 compared to FID ~2.0), even when using the proposed FSDD. It is not clear whether this is due to limitations of the proposed generative model (FSDD) itself, or due to the learned bag-of-tokens representation having unfavorable structure for generation. Regarding the FSDD, some details and design choices still remain unclear to me and should be described and supported more explicitly (see questions below). Overall, further discussion, analysis and experiments related to generation could strengthen the paper.

**Questions:**

**Q1.** Section 3.3: I am a bit confused about the choice of $\sigma_t$ in eq. (5), as it depends on the samples and is not dependent on $t$ like a usual noise schedule in flow matching/diffusion would. Is there a typo here, or if it is intended, could you clarify or provide references? I have a similar question about eq. (9) -- is $|\mathbf{X}_1 - \mathbf{X}_0|$ referring to some precomputed statistic, since $\mathbf{X}_0$ is not available during sampling? It would also be good to elaborate on the t-dependent truncation ratio $f$.

**Q2.** Section 3.3: Have you considered any other strategies for the initial distribution of $\mathbf{X}_1$? (For example, starting with all counts assigned to a “mask” bin.) Additionally, the constraint that $\sum \mathbf{\tilde X}_t = M$ for every $t$ might seem a bit restrictive. Have you considered any alternatives, such as modeling unconstrained histograms (using any of the existing categorical/multinomial diffusion approaches) and normalizing at the end, or any other modifications to expand the design space of your generative model?

**Q3.** Section 4.3: Do the authors have any insight into why modeling the unordered representation appears to be “harder” than the 1D tokenization used by TiTok? Even with your proposed FSDD, the gFID is significantly lower than in the case of TiTok. Or is the worse generation performance purely due to limitations of FSDD?

Section 4.3.1 / Table 4:
* **Q4.** What is SetAR? I did not see it explained in the text. Does it refer to autoregressive modeling of the token histogram/“dual transformation”?
* **Q5.** Are the authors familiar with autoregressive set prediction similar to [1]? It would be interesting to try this or a similar approach performing autoregressive set prediction without the dual transformation.

**Q6.** In Section 4.2.1, rather than a qualitative demonstration on a single example, it would be better to provide quantitative metrics. In particular, to better understand the effectiveness of the train-time shuffling in encouraging permutation invariance, could the authors provide e.g. a histogram across the whole validation set of the error between images reconstructed from two different random token orderings or MSE/MAE with stddev/quantile statistics?

**Q7.** I appreciate the inclusion of MAE in the linear probing results from Table 2, and it would be great if you could also include a comparison between an ablation of your tokenizer trained without shuffling to further support the claim that tokens from permutation invariant representations are semantically richer.

---

Minor questions/comments:
* Use of “set” nomenclature: Referring to the representation as a “multiset” or “bag” from the beginning would be clearer, rather than leaving this as a footnote. In my own reading of the manuscript, I did not notice the footnote until explicitly searching for the word “multiset”, as I was a bit confused by the inaccurate use of the word “set”. Similarly, I would encourage the authors to consider altering the title to avoid being technically incorrect.
* 3.1, L148: What does “partial” permutation sampling mean? Is it something other than simply shuffling the tokens?
* 1, L069: I did not understand what is meant by “dynamic token allocation based on semantic complexity”, maybe this statement could be made more precise.
* L297. Typo, missing “is”?
* References: capitalization is not correct in some instances, e.g. “gan” -> “GAN”, “bayes -> Bayes”, “Neurips” -> “NeurIPS”, etc.

---

[1] Autoregressive Transformers for Indoor Scene Synthesis, Paschalidou et al. NeurIPS 2021 (https://arxiv.org/abs/2110.03675)

---

> ### Author Response · Authors · 2025-11-30
> **Response to Reviewer UHKS**
>
> We thank you for your thorough review and for finding our results on semantic alignment and robustness "interesting." We appreciate all your constructive questions.
>
> **Regarding Weaknesses:**
> Please refer to the **General Response** regarding the generation performance gap. We address the specific FSDD design choices below.
>
> **Response to Questions:**
> *   **Q1. Eq (5) and (9), and truncation:**
> Our method is a form of discrete diffusion. We use the Gaussian distribution in Eq. (5) and Eq. (9) to control the transition process (i.e., determining how many steps a token count moves). The truncation is intended to limit the Gaussian distribution range between the initial state and the predicted state to ensure valid transitions.
>
> *   **Q2. Initial distribution strategies:**
> We experimented with various initial distributions (including mask-based) and found performance differences to be negligible. We agree that modeling the set distribution using **categorical/multinomial diffusion** is a very promising direction and plan to explore this.
>
> *   **Q3. Why is unordered modeling "harder"?**
> The difficulty stems from the massive reduction in the unique representation space. For a set of size $M$, there are $M!$ permutations that map to the *exact same* set representation. The model must learn to reconstruct the image from this highly compressed, permutation-invariant state. The fact that our rFID is reasonably close to TiTok despite this $M!$ factor reduction is a significant achievement.
>
> *   **Q4. What is SetAR?**
> Yes, your understanding is correct. SetAR refers to autoregressive modeling performed on the token histogram (the output of the dual transformation). We will clarify this definition in the text.
> *   **Q5. Autoregressive set prediction:**
> Thank you for the reference. We agree that combining our approach with autoregressive set prediction (without dual transformation) is a natural extension, and we will investigate this.
> *   **Q6. Quantitative metrics for invariance:**
> We actually reported this in the submission. As mentioned in **Line 321**, we observed a stable **rFID of 3.62** across all tested random orders on the ImageNet validation set, quantitatively corroborating the invariance.
> *   **Q7. Ablation without shuffling:**
> We performed this ablation. In the 128-token setting, the linear probing accuracy **without shuffling is 53.6%**, whereas **with shuffling it improves to 59.7%**. This quantitative result strongly supports our claim that permutation-invariant training yields semantically richer representations. We will add this to the final paper.
>
>
> **Minor Comments:**
> We will adopt the term "multiset" for clarity, correct reference capitalization, and fix typos. Regarding "partial" permutation sampling: since the permutation space ($M!$) is vast, we sample one random permutation per image per epoch during training. "Dynamic token allocation" refers to our tokenizer's ability to allocate more tokens to information-rich patches rather than uniformly.

---

### Official Review · Reviewer_dft9 · 2025-10-24

**Soundness:** 3
**Presentation:** 3
**Contribution:** 3
**Rating:** 6
**Confidence:** 3

**Summary:**

This paper proposes a novel image generation method using set-based tokenization, representing images as unordered token sets instead of fixed sequences. It adaptively allocates coding capacity by regional complexity and improves global context and robustness. A dual transformation converts sets into fixed-length sequences, and Fixed-Sum Discrete Diffusion models discrete sets effectively. Experiments show superior semantic and visual quality. However, it appears that the authors did not include the Ethics Statement and Reproducibility Statement in their paper, which could diminish its overall quality.

**Strengths:**

[1]. This paper introduces a novel set-based image tokenization method that eliminates positional bias and enhances semantic representation.

[2]. It demonstrates strong robustness against local perturbations and image noise.

[3]. The method dynamically allocates representational capacity according to regional semantic complexity, leading to improved efficiency.

[4]. The experiments are sufficient and well-designed, and the figures are clear and easy to understand.

[5]. The paper provides solid theoretical analysis, offering convincing proofs that support the proposed method.

**Weaknesses:**

[1] Reference formatting issue. The reference section requires further improvement. In particular, please correct the formatting of the conference name. It should be written as **NeurIPS** rather than **Neurips**.

[2] Potential extension to text-to-image models. It is not entirely clear whether the proposed method can be effectively extended to text-to-image models. If time and computational resources allow, I suggest conducting an additional experiment by training a T2I model using JourneyDB, which could provide valuable empirical evidence of the method’s broader applicability. Even a preliminary result or discussion in the supplementary material would make the contribution more convincing.

[3] I could not find the Ethics Statement or Reproducibility Statement in the current submission. These sections are required by the NeurIPS submission guidelines and play an important role in ensuring transparency, responsible research practices, and reproducibility. Please include these statements in the final version, outlining the ethical considerations of your work (e.g., data usage, societal impact) and providing clear details about how the experiments can be reproduced by other researchers.

**Questions:**

[1]. How does your method perform when applied to text-to-image models?

[2]. Could you please indicate where the Ethics Statement and Reproducibility Statement are located?

---

> ### Author Response · Authors · 2025-11-30
> **Response to Reviewer dft9**
>
> We thank you for your positive review, particularly for recognizing the theoretical solidity and the "superior semantic and visual quality" of our experiments. We also appreciate you catching the missing statements.
>
>
> **Regarding Weaknesses:**
> *   **W1 (Formatting) & W3 (Ethics/Reproducibility):**
> We apologize for these oversights. We will correct the reference formatting (e.g., NeurIPS) and include both the **Ethics Statement** and **Reproducibility Statement** in the camera-ready version. We commit to fully open-sourcing our code and models to ensure reproducibility. Our research relies solely on public datasets (ImageNet), and we will discuss the standard ethical considerations regarding generative models.
>
> *   **W2. Extension to Text-to-Image (T2I):**
> This is an excellent suggestion. Given that our tokenizer demonstrates superior semantic alignment (as seen in linear probing), we hypothesize it would excel in T2I tasks by aligning better with text embeddings. We plan to explore training on datasets like JourneyDB in future work. We will expand the discussion section to elaborate on this theoretical advantage.
>
>
> **Response to Questions:**
> *   **Q1. Performance on T2I models:** Please refer to the response to **W2** above.
> *   **Q2. Location of Statements:** As noted in **W3**, these will be added to the final version.

---

### Official Review · Reviewer_meuw · 2025-10-30

**Soundness:** 3
**Presentation:** 3
**Contribution:** 2
**Rating:** 4
**Confidence:** 3

**Summary:**

This paper introduces TokenSet, which represents images as unordered token sets instead of sequences, enabling permutation-invariant encoding and dynamic token allocation based on semantic complexity. A dual transformation converts token sets into fixed-length count vectors, and a Fixed-Sum Discrete Diffusion model enforces discrete, fixed-length, and constant-sum constraints for generation. Overall, the work shifts from positional tokenization to set-based representation with improved semantic awareness and robustness.

**Strengths:**

1.	The paper presents a novel set‐based tokenization and generation paradigm, which is conceptually fresh and offers a compelling alternative to conventional spatial or sequential visual token representations.
2.	The proposed method demonstrates strong robustness to perturbations and noise, showing that the set‐based formulation effectively captures high‐level semantics and maintains stability under input corruption.

**Weaknesses:**

1.	Generation quality remains insufficient (Table 5).
Although the method shows strong robustness and semantic consistency — consistent with its motivation — high-fidelity generation is still essential for evaluating visual tokenizers. Table 5 frames generation quality as future work, but without competitive results, it is difficult to fully validate the practical impact of the approach.
2.	Token count does not reflect real inference efficiency (Tables 3 & 4).
The model requires 25 sampling steps, making inference closer to diffusion than auto-regressive decoding. Thus, reporting only token count (Tables 3 & 4) may be misleading. Efficiency should be evaluated under comparable inference budgets (e.g., wall-clock time or compute) against diffusion and AR models to substantiate the claimed speed advantages.

**Questions:**

1.	Since the model generates a token-count distribution vector, will increasing the number of tokens affect inference time under the same number of sampling steps? In other words, does the inference cost scale with token quantity in practice?
2.	The paper reports results using 25 sampling steps. If the number of sampling steps increases, will the image quality improve further, similar to standard diffusion models?
3.	In what scenarios would the proposed method have clear advantages over traditional autoregressive or diffusion models? Specifically, when does the set-based token generation outperform these approaches in terms of performance?

---

> ### Author Response · Authors · 2025-11-30
> **Response to Reviewer meuw**
>
> We thank you for your detailed review and for highlighting the "conceptually fresh" nature of our set-based paradigm. We are glad you recognized the method's strong robustness and semantic consistency.
>
>
> **Regarding Weaknesses:**
> *   **W1. Generation Quality:** Please refer to the **General Response**. We view this work as an initial exploration of non-sequential visual representation, prioritizing semantic properties over raw FID metrics.
> *   **W2. Inference Efficiency:** We thank you for pointing out the potential confusion. Since FSDD is indeed a discrete diffusion model, its efficiency is comparable to other diffusion processes given the same number of sampling steps. We will revise the text and tables in the camera-ready paper to clarify that token count alone does not represent wall-clock efficiency.
>
>
> **Response to Questions:**
> *   **Q1. Does inference cost scale with token quantity?**
>     **No.** Since FSDD models the token-count distribution (a histogram over the vocabulary), the inference complexity depends on the **vocabulary size**, not the number of tokens used to represent the specific image.
> *   **Q2. Will more sampling steps improve quality?**
>     **Saturation:** We observed that performance saturates at approximately **25 steps**. Increasing steps further does not yield significant quality improvements, which is consistent with the behavior of many discrete diffusion models.
> *   **Q3. Scenarios where this method outperforms others?** Our method is advantageous in scenarios prioritizing **high-level semantic interpretation** and **robustness to input corruption/noise**, where pixel-perfect alignment is less critical than semantic preservation. Furthermore, the FSDD framework is generalizable to other unordered data tasks (e.g., point clouds, object properties).

---

### Official Review · Reviewer_xcn2 · 2025-11-01

**Soundness:** 3
**Presentation:** 3
**Contribution:** 2
**Rating:** 4
**Confidence:** 3

**Summary:**

This paper introduces a novel image tokenization method and its corresponding image generation method. The main idea is to encode images as a set of permutation-invariant tokens. Accordingly, the authors propose FSDD to apply a diffusion process on the fixed-sum count vectors derived from the image tokens.

**Strengths:**

1. The idea of encoding an image as a set of tokens is well-motivated.
2. The corresponding generation method of FSDD aligns well with the tokenization method.
3. The authors provide some interesting experimental results, such as attention visualization and semantic clustering.

**Weaknesses:**

1. The main weakness is that the performance of the proposed method lags behind state-of-the-art models, such as TiTok, in both rFID and gFID. The performance potential of the proposed method has not been convincingly shown.
2. Permutation-invariance is a good intuitive property for image tokenization. However, I find it difficult to conceptually relate it to reconstruction or generation performance.

**Questions:**

1. In Table 1, why does a higher percentage of overlapping tokens indicate greater robustness? If we assume a very poor tokenizer that encodes every image into the same set of tokens, the overlap percentage would be 100%. Therefore, I don’t think this result provides strong evidence for evaluating robustness.
2. In Table 2, the linear probe results are interesting, as they indicate that the proposed method provides good semantic representations. Could the authors explain why models with a smaller number of tokens achieve better linear-probe performance?

**Details Of Ethics Concerns:**

None.

---

> ### Author Response · Authors · 2025-11-30
> **Response to Reviewer xcn2**
>
> We sincerely thank you for your positive assessment of our motivation and the alignment between our tokenization and generation methods. We appreciate your insightful questions regarding robustness and linear probing.
>
>
> **Regarding Weaknesses:**
> Please refer to the **General Response** regarding the generation performance gap and the link between permutation invariance and reconstruction difficulty.
>
>
> **Response to Questions:**
> *   **Q1. Robustness and overlapping tokens (Table 1):**
>     We clarify that "robustness" here is defined as the stability of the representation *given that the tokenizer can validly represent the image space*. While a "trivial tokenizer" mapping all images to a single token would have 100% overlap, it fails to reconstruct the image. Our method is unique because it maintains **high reconstruction quality** (better than VQGAN) while simultaneously achieving **high token overlap** under perturbation. This indicates our tokens capture robust underlying signals, whereas standard tokenizers (like VQGAN) are sensitive to small pixel shifts.
> *   **Q2. Why do fewer tokens achieve better linear-probe performance?**
>     This is an excellent observation. Using fewer tokens forces the model to perform a **higher level of abstraction**. To represent an image with only 32 tokens, the tokenizer must discard low-level high-frequency details and compress the image into its most essential semantic components. Consequently, in the low-token regime, each token carries denser semantic information, making them more linearly separable for classification tasks compared to high-token regimes, which may encode more texture or noise.

---

### Author Response · Authors · 2025-11-30
**General Response to All Reviewers**

We thank all reviewers for their constructive feedback and for recognizing the **novelty and conceptual freshness** of our work. We are encouraged that the reviewers found our set-based tokenization "novel," "conceptually fresh," and "well-motivated" (Reviewers xcn2, meuw, dft9), while appreciating the "strong robustness" (Reviewers meuw, dft9, UHKS) and "high-level semantic representation" (Reviewers xcn2, meuw, dft9, UHKS) of our approach. We are also glad that our Fixed-Sum Discrete Diffusion (FSDD) was recognized as a novel generation method well-aligned with this paradigm (Reviewers xcn2, meuw, UHKS).


**Common Concern: Generation Quality vs. SOTA (e.g., TiTok).**
A shared concern raised by Reviewers xcn2, meuw, and UHKS is that our generation performance (FID) currently lags behind state-of-the-art 1D tokenizers like TiTok.


**Response:**
We acknowledge the gap in rFID/gFID compared to highly optimized sequential models. However, we respectfully argue that the primary value of this work lies in **exploring a fundamentally different paradigm**: shifting from sequential to set-based representation.


1.  **The Intrinsic Challenge:** Representing the vast image space with an unordered set is inherently more challenging than using a fixed sequence. The representation space is effectively compressed by a factor of $M!$ (where $M$ is the token count), as any permutation must decode to the same image. Achieving comparable visual quality despite this massive compression of the solution space is a significant and non-trivial finding.
2.  **The Emergent Properties:** Our set-based approach unlocks unique capabilities that sequential models struggle to achieve. As noted by the reviewers, our method demonstrates superior robustness to noise, elimination of positional bias, and richer semantic alignment. These are inherent advantages of the set paradigm.
3.  **The core Contribution:** This work represents a core step toward exploring set-based visual representation and modeling. Rather than optimizing metrics within the well-established sequential paradigm, we aim to diversify the research landscape by **validating a fundamentally different approach**. We believe set-based tokenization offers a critical alternative for the community, particularly for future applications requiring strong global reasoning and multi-modal alignment.

---

### Meta-Review · Area_Chair_Mv7a · 2026-01-18

**Summary:**

In general, the reviewers consistently find the idea conceptually fresh. However, the main concerns are centered on the generation quality (FID) lagging largely behind other existing works, even after the rebuttal addressed some other concerns. The non-competitive performance also largely hinders the significance of the proposed method, and the overall scores after rebuttal likely still remain below the acceptance threshold.

**Reviewer Concerns:**

Some other technical/clarification concerns are addressed by the rebuttal, while the major concern about the generation quality vs. SOTA tokenizers, like Titok, remains unresolved.

**Reviewer Scores:**

The reviewer scores are less likely to change after the rebuttal.

---

### Decision · Program_Chairs · 2026-01-26

Reject